# Unraveling DNA Repair Processes In Vivo: Insights from Zebrafish Studies

**DOI:** 10.3390/ijms241713120

**Published:** 2023-08-23

**Authors:** Unbeom Shin, Yoonsung Lee

**Affiliations:** 1School of Life Sciences, Ulsan National Institute of Science and Technology (UNIST), Ulsan 44919, Republic of Korea; rkwhr0126@unist.ac.kr; 2Clinical Research Institute, Kyung Hee University Hospital at Gangdong, School of Medicine, Kyung Hee University, Seoul 05278, Republic of Korea

**Keywords:** zebrafish, DNA repair, genome stability, neurogenesis, hematopoiesis, germ cell development, cancer, aging

## Abstract

The critical role of the DNA repair system in preserving the health and survival of living organisms is widely recognized as dysfunction within this system can result in a broad range of severe conditions, including neurodegenerative diseases, blood disorders, infertility, and cancer. Despite comprehensive research on the molecular and cellular mechanisms of DNA repair pathways, there remains a significant knowledge gap concerning these processes at an organismal level. The teleost zebrafish has emerged as a powerful model organism for investigating these intricate DNA repair mechanisms. Their utility arises from a combination of their well-characterized genomic information, the ability to visualize specific phenotype outcomes in distinct cells and tissues, and the availability of diverse genetic experimental approaches. In this review, we provide an in-depth overview of recent advancements in our understanding of the in vivo roles of DNA repair pathways. We cover a variety of critical biological processes including neurogenesis, hematopoiesis, germ cell development, tumorigenesis, and aging, with a specific emphasis on findings obtained from the use of zebrafish as a model system. Our comprehensive review highlights the importance of zebrafish in enhancing our understanding of the functions of DNA repair systems at the organismal level and paves the way for future investigations in this field.

## 1. Introduction

DNA damage is a ubiquitous and potentially detrimental event that can trigger a range of cellular processes from alterations in cell cycle progression and transcriptional programs to cell death [1]. This damage can arise from a variety of sources, including external radiation, oxidative chemicals, DNA replication stress, or errors in cellular metabolism [2]. To protect the integrity of their genome against these numerous threats, human cells have evolved diverse DNA repair pathways capable of addressing different types of damage. These repair pathways include base excision repair (BER), nucleotide excision repair (NER), mismatch repair (MMR), and DNA double-strand break repair (DSBR) such as homologous recombination (HR) and non-homologous end-joining (NHEJ) (Figure 1) [3,4]. Each of these pathways play a distinct and vital role in maintaining genomic stability. As such, any failure in these repair mechanisms can give rise to various diseases, most notably cancer and developmental disorders [5,6]. Given the clear links between DNA repair deficits and disease, unraveling the functions of the genes involved in these DNA repair pathways is crucial for understanding the pathological mechanisms of the disease and devising therapeutic strategies.

While significant progress has been made in deciphering the molecular and cellular mechanisms underlying various DNA repair pathways, the roles that these systems play within living organisms remain largely undefined. Over the past several decades, studies employing mouse models, specifically those with loss-of-function approaches, have demonstrated the importance of DNA repair pathways in vertebrate development [7]. The targeted investigation of specific genes has further enhanced our understanding, revealing their distinct roles in essential biological processes such as neurogenesis, hematopoiesis, and germ cell development [8,9,10,11,12,13]. However, despite these advances, the majority of genes involved in DNA repair have not been thoroughly examined using the knockout (KO) mice system, primarily due to the early embryonic lethality [14,15,16,17,18,19]. This challenge highlights the necessity of alternative research methodologies, such as the generation of tissue-specific or conditional KO animals [15,20]. However, the application of these methods brings its own challenge, namely that they are time-consuming, labor-intensive, and require significant technical expertise.

Teleost zebrafish have emerged as a highly appealing model organism for molecular genetics and human disease research, largely due to their diverse availability of genetic experimental approaches [21]. The advent of CRISPR mutagenesis has opened up unprecedented opportunities in genetic manipulation, allowing for the efficient generation of KO animals and subsequent phenotypic observations at various developmental stages [22]. A particularly attractive aspect of zebrafish is the rapid developmental process during embryogenesis, enabling researchers to investigate the functions of genes whose deficiency causes early lethality in mammals. Remarkably, these functions can be observed in KO zebrafish even beyond the initial developmental stage [23,24]. This capability proves invaluable when studying the genes involved in DNA repair processes, which are generally critical during early embryogenesis [25]. Moreover, the substantial genetic and physiological similarities between zebrafish and humans enable findings in zebrafish to be extrapolated to human biology, thereby advancing our understanding of human disease pathology [26].

In this review, we will explore the advantages of zebrafish as an alternative animal model for studying DNA repair pathways. We aim to provide a comprehensive overview of the genetic characteristics and functional roles of DNA repair-related genes within various biological contexts, such as neurogenesis, hematopoiesis, germ cell development, tumorigenesis, and aging. This will involve an in-depth examination of how specific DNA repair mechanisms are implicated in these processes, as well as the potential consequences when these mechanisms fail or become dysregulated. Our ultimate goal is to highlight the substantial contribution that research using zebrafish can make to our understanding of DNA repair processes and their potential roles in related diseases. By providing a broader view of the interplay between DNA repair and various biological processes, we hope to pave the way for novel therapeutic strategies to treat diseases associated with DNA repair deficiencies.

## 2. Zebrafish as a Genetic Model for Studying DNA Repair Genes

Zebrafish, an excellent model organism, boast a considerable genomic resemblance to humans, with studies suggesting that approximately 70 percent of its genome parallels human genomic information [27]. This significant overlap underscores the relevance of zebrafish to human biology and amplifies their value as a model for studying genetic phenomena, including DNA repair mechanisms. A more detailed examination reveals that this substantial genomic conservation extends to most DNA repair genes in zebrafish, indicating that their DNA repair systems are similar to those found in other vertebrates, including humans [28,29].

Zebrafish present a unique advantage by allowing for observable KO effects for essential genes related to DNA repair processes even beyond early development, an attribute often lacking in KO mice models for these genes which generally do not survive past prenatal stages (Table 1). This edge is attributed to the relatively rapid developmental progression of zebrafish and the presence of durable maternal mRNA effects in their embryos. Unlike in mice models where most maternal mRNAs degrade prior to the two-cell stage (~24 h post fertilization (hpf)), zebrafish maternal mRNA remains intact until the gastrulation stage (~6 hpf) [30,31]. Multiple studies have provided evidence that specific DNA repair genes, which lead to embryonic lethality in mouse models when knocked out, can persist in zebrafish throughout embryonic stages and even into adulthood. For instance, unlike mice where the KO of *Ddb1*, *Atad5*, *Pcna*, and *Topbp1* leads to early embryonic lethality, zygotic mutations of those genes in zebrafish have allowed them to survive during embryonic stages [29,32,33]. Similarly, *telo2*, *mre11*, and *ino80* KO zebrafish showed viability in the larval stage [29]. Moreover, previous studies have shown that loss of *blm*, *brca2*, *rad51*, *palb2*, *rtel1*, and *xrcc1* also yields viable zebrafish, even in the adult stages [15,17,19,29,34,35,36,37].

Zebrafish provide several advantages for studying genes during embryogenesis due to key factors such as their high fecundity, developmental transparency, and the use of specialized transgenic reporter lines that allow the visualization of specific tissues and organs [38,39,40]. Additionally, various genetic modification tools are available for conducting these intricate genetic studies. These include both forward and reverse genetic methodologies, with techniques such as N-ethyl-N-nitrosourea (ENU)-mediated random mutagenesis, morpholino injection for transient gene knock-down, and gene editing through the CRISPR/Cas9 system for target gene KO [41,42,43,44,45]. Furthermore, recent advancements in multiplexed mutagenesis via the CRISPR/Cas9 system now facilitate the generation of large-scale target gene mutants of zebrafish [46]. Therefore, for in-depth exploration of DNA repair gene functions during embryogenesis and throughout the entire lifespan, zebrafish stand as potent and flexible model organisms.

**Table 1 ijms-24-13120-t001:** Mouse KO studies showing embryonic lethality. This table lists various DNA repair genes whose KO in mice resulted in embryonic lethality. Details such as the specific genes, stages at which lethality occurs, and references to original studies are provided. This lethality has limited further investigations of these genes in mouse models.

DNA Repair Process	Gene	Timepoint of Lethality	References
Checkpoint	*Atr*	E7.5	Brown and Baltimore, 2000 [47]
	*Chk1*	E7.5	Takai et al., 2000 [48]
	*Ino80*	E7.5	Qiu et al., 2016 [18]
	*Topbp1*	E8.5	Jeon et al., 2019 [49]
	*Men1*	E8.5	Crabtree et al., 2001 [50]
	*Smg1*	E8.5	Roberts et al., 2013 [51]
	*Bard1*	E8.5	McCarthy et al., 2003 [52]
	*Telo2*	E13.5	Takai et al., 2007 [35]
	*Rad17*	E13.5	Budzowska et al., 2004 [53]
Nucleotide excision repair	*Ddb1*	E12.5	Cang et al., 2006 [15]
Non-homologous end joining	*Lig4*	E15.5	Barnes et al., 1998 [54]
Homologous recombination	*Xrcc1*	E7.5	Tebbs et al., 1999 [36]
	*Rad51*	E8.5	Lim and Hasty, 1996 [17]
	*Palb2*	E8.5	Bowman-Colin et al., 2013 [55]
	*Mre11*	E9.5	Buis et al., 2008 [14]
Telomere maintenance	*Rtel*	E11.5	Ding et al., 2004 [16]
DNA replication	*Atad5*	Embryonic lethal	Bell et al., 2011 [56]
	*Pcna*	Embryonic lethal	Roa et al., 2008 [57]

## 3. The Roles of Genes Related to DNA Repair Processes in Zebrafish Development 

Taking advantage of the zebrafish, extensive characterization of DNA repair-related genes has been conducted both at individual and multiple gene levels. These studies have not only confirmed previously described gene functions but also uncovered the roles of DNA repair pathways in specific cell types. In the following sections, we will focus on findings related to the functions of DNA repair genes in zebrafish, with particular emphasis on five key aspects: neurogenesis, hematopoiesis, germ cell development, tumorigenesis, and aging (Figure 2). This review showcases how these studies have significantly expanded our understanding of DNA repair pathways and their multifaceted roles in different biological processes.

### 3.1. Neurogenesis

During development, neural cells undergo rapid proliferation and DNA replication. This increased activity often heightens the potential for DNA damage due to replication errors or environmental factors. Failure to maintain genomic integrity, particularly due to the inability to repair these DNA damages efficiently, has been associated with neurodegenerative disorders [58]. To understand the impact of DNA repair genes on neurogenesis, various studies have leveraged the advantage of zebrafish as a model organism. By employing gene knock-down and KO strategies in zebrafish, researchers have been able to reveal valuable insights into the roles of DNA repair genes on several stages of neurogenesis, such as neural cell proliferation, differentiation, and migration.

For example, knock-down of the BER component *ogg1*, which recognizes oxidative DNA damage and initiates BER pathway, leads to brain malformations in early embryonic stages [59]. Several cellular events were involved in the development of brain defects in the absence of *ogg1* including escalated apoptotic events, diminished cell proliferation, and distorted axon distribution, accompanied by DNA damage accumulation. Similarly, knock-down of *apex1*, another gene involved in the BER pathway, results in abnormal early brain formation with elevated levels of reactive oxygen species (ROS). Loss of *apex1* resulted in a reduction in the expression of several genes related to the BER pathways including *ogg1* and *polb* [60]. Furthermore, the expression of several key brain transcription factors like *fezf2*, *otx2*, *egr2a*, and *pax2a* was disrupted in the absence of *apex1*, leading to abnormal brain formation. In addition, several variants of the *LIG3*, also part of the BER pathway, were identified in patients showing aberrant mitochondrial functions and neurological abnormalities [61]. Knock-down of *lig3* leads to a cerebellar hypoplasia phenotype with an incompletely developed brain in zebrafish embryos. Collectively, these findings emphasize the requirement of the BER pathway for proper neurogenesis.

Additional genetic studies have demonstrated the importance of maintaining genomic integrity during neural development. In a large-scale screening of zebrafish mutants for DNA repair pathway genes, it was found that the deletion of *pcna*, the eukaryotic sliding clamp, leads to an abnormal anterior head structure with a reduced size of brain from 2 days post fertilization (dpf) [29]. Separately, PCNA unloader *atad5a* mutants were found to exhibit abnormal brain morphology at a later stage of 5 dpf, with induced apoptosis in the brain [29]. In addition, the deletion of *banp*, a regulator of p53 activity, results in eye development defects by inducing replication stress and p53-mediated DNA damage response-induced apoptosis [62]. A loss of *baz1b*, which is critical for chromatin remodeling and DNA repair and replication, leads to the aberrant formation of neural crest cells during early embryogenesis, while anxiety-associated behavior phenotypes were observed during adult stages [63]. Moreover, a mutation of the cell cycle gene regulator *rbbp4* results in a reduced size of the brain and eyes due to mitotic arrest and the subsequent apoptosis of neural progenitor cells, leading to the loss of differentiated neurons [64]. In another study, the deletion of *sgo1*, a cohesin regulator, is implicated in developmental defects, including abnormal eye morphology [65]. Additionally, the loss of *osgep* and *tpkrb*, tRNA processing factors which have been identified to regulate the maintenance of genomic stability, leads to microcephalic phenotypes with induced apoptosis in zebrafish [66]. Taken together, these findings demonstrate that both proper proliferation and maintenance of genome stability are critical for brain and neural development.

Mutations in other DNA repair genes further support the importance of DNA repair pathways in neurogenesis. Mutations in the NER gene *ddb1*, induced by ENU random mutagenesis, lead to a smaller and abnormally shaped brain as well as the malformation of dopaminergic neurons, along with the dysregulation of cell cycle genes [33]. Additionally, the characterization of the telomere protector *trf2* reveals its role in activating neuronal genes during zebrafish embryogenesis [67]. The knock-down of *trf2* disrupted the expression network of neuronal genes. In summary, these findings provide insights into the biological mechanisms underlying neurodevelopment and associated disorders, underlining the potential of using zebrafish to further explore the complex relationship between DNA repair and neurogenesis.

### 3.2. Hematopoiesis

Numerous studies have shown that mutations in genes related to DNA repair are frequently identified in patients with blood disorders, emphasizing the importance of these genes in hematopoietic systems [68,69,70]. Zebrafish carry hematopoietic genes and regulatory networks that are evolutionarily conserved, and all mature blood lineages and their corresponding regulatory networks have been identified and characterized in zebrafish to date [71]. Diverse genetic studies focusing on DNA repair genes have highlighted the essential role of genomic integrity in various hematopoietic processes in zebrafish. One gene of interest is the small ubiquitin-related modifier (SUMO)-targeted E3 ligase, RNF4. RNF4 is primarily involved in DNA damage repair and chromatin regulation, contributing significantly to maintaining genomic integrity and the loss of *rnf4* results in defects in embryonic myeloid development in zebrafish [72]. RNF4 regulates granulopoiesis via the regulation of DNA methyltransferase 1 (DNMT1) and the CCAAT/enhancer-binding protein α (C/EBPα) axis.

Hematopoietic stem and progenitor cells (HSPCs) are multipotent stem cells capable of generating all blood lineages for life. Interestingly, various functional studies in zebrafish have shown that genes related to DNA repair and the maintenance of genome integrity are associated with the development of HSPCs. For example, the mutation of *topbp1*, a gene involved in DNA damage checkpoint activation, significantly affects the survival of HSPC populations during the embryonic stage in zebrafish [32]. Without normal functions of Topbp1, DNA damage is accumulated, eventually activating p53-depdent apoptosis in the HSPCs. Furthermore, null mutations of either *pcna* or *atad5a* in zebrafish result in a reduced number of HSPCs due to blockade of proliferation as well as induced cell death in the blood precursors, while homozygous mutants of *ddb1* showed a loss of HSPCs solely caused by reduced proliferation [29]. Similarly, the absence of the cohesin subunits *rad21* and *smc3* results in a reduced expression of *runx1*, a marker of HSPCs, emphasizing the fundamental role of genomic integrity in normal blood development [73]. These findings underline the necessity of proper DNA replication and maintenance of genomic integrity for both the formation and maturation of normal HSPCs.

HSPCs, originating during the early developmental stage, establish themselves within the kidney marrow in zebrafish, regulating and maintaining blood homeostasis through the organism’s life, from the larval stages to adulthood. Crucially, genes associated with DNA repair and genomic integrity not only play a vital role in the initial stages of hematopoiesis during embryogenesis, but also are essential for preserving the normal functions of blood cells in later developmental stages, extending to adulthood. For instance, a loss of the HR gene *rad51* results in a noticeable decrease in the blood cell population within the kidney marrow of adult zebrafish, possibly due to the p53-dependent apoptosis of embryonic HSPCs [74]. Similarly, mutations of the *tonsl* gene, crucial for DSBR, lead to diminished neutrophil counts during the larval stage, suggesting a vital role for DSBR in maintaining blood homeostasis [75]. Moreover, adult zebrafish with mutations in the DNA damage checkpoint kinase *atm* gene exhibit features resembling those in human ataxia-telangiectasia. These include immune deficiencies that cause widespread infections and lead to tumor formation [76].

Interestingly, genes related to hematologic malignancies also shed light on the role of DNA repair systems in blood disorders [77]. Numerous variants in DEAD-box Helicase 41 (DDX41) have been identified in diverse blood disorders including leukemia. In zebrafish, a loss of *ddx41* results in diminished proliferation and defective differentiation of erythroid progenitors, likely triggered by cell cycle arrest mediated by ATM and ATR [78]. Moreover, an overexpression of oncogenic MYCN leads to a significant presence of massive immature hematopoietic stem cells in the kidney marrow, accompanied by the downregulation of the BER, HR, and MMR pathways [79]. This highlights the importance of DNA repair pathways in hematologic malignancies. Overall, the comprehensive characterization of DNA repair genes using zebrafish has elucidated their roles in general blood development, highlighting their potential implications in blood disorders and malignancies.

### 3.3. Germ Cell Development

Various mutations identified in human DNA repair genes specifically impact germ cell function and development. For instance, mutations in Fanconi anemia (FA) genes, responsible for DNA cross-link repair, frequently lead to FA, a condition that is characterized by a range of symptoms including bone marrow failure, an increased risk of leukemia, hypogonadism, and infertility [80]. Zebrafish models have been instrumental in studying the complex relationship between DNA repair genes and germ cell development. Intriguingly, studies in zebrafish have demonstrated that mutations in FA genes can lead to defects in germ cell development, resulting in female-to-male sex reversal. Adult homozygous *fancl* zebrafish mutants are exclusively males, and the germ cells in the gonads of these mutants undergo apoptosis mediated by p53. This suggests that the observed sex reversal results from a failure in oocyte generation due to germ cell apoptosis [81]. Similarly, mutations in the HR/FA gene *brca2* (*fancd1*) also cause sex reversal from female to male in adult stages of zebrafish. Juvenile *brca2* mutants fail to develop ovaries, and meiosis is arrested in the testes. Like *fancl* mutants, the defect in *brca2* mutants is associated with p53 function, as evidenced by the phenotype of *p53*; *brca2* double mutants [82].

Moreover, a recent comprehensive characterization of 19 FA genes in zebrafish revealed that all adult homozygous mutants exhibit features of female-to-male sex reversal underscoring the essential roles of DNA cross-link repair in germ cell development [37]. Further investigation demonstrated that mature spermatozoa were absent in the testes of *brca2* and *fancj* mutants. Meanwhile, co-mutation of *p53* with *fancp* knockout in adult zebrafish rescued their reversal phenotypes. Bloom syndrome, which is caused by mutations in BLM helicase for HR and DNA cross-link repair, also affects germ cells, leading to infertility in male patients [83]. A depletion of *blm* in zebrafish leads to a bias towards male phenotypes with reduced fertility. In the testes of *blm* mutants, spermatocytes are arrested in meiotic phase I, suggesting that infertility in *blm* mutants is caused by meiotic arrest of spermatocytes [84]. Taken together, these observations highlight the crucial role of the FA/HR pathway in repairing DNA cross-link for proper germ cell development.

In addition to FA/HR genes, mutations in other genes have also been implicated in the sex reversal phenotype. Characterization of the mutants of the MMR gene *mlh* demonstrated that *mlh* is involved in the maintenance of germ cells in both males and females. A loss of *mlh* in male zebrafish leads to the arrest of spermatogenesis at metaphase I resulting in infertility, while female *mlh* mutants are fertile but their progeny exhibit lethality with aneuploidy [85]. Homozygous mutants of the cohesin subunit *rad21l1* exhibit a female-to-male reversal phenotype. Interestingly, double mutation of *rad21l1* with p53 restored the reversal phenotype, further supporting the importance of p53-mediated apoptosis for the survival of germ cells [86]. Additionally, male-biased effects were observed in *rad54l* and *rtel1* mutants through a large-scale screening of DNA repair genes [29]. Collectively, the characterization of DNA repair genes in zebrafish has revealed their crucial role in the maintenance of germ cells, highlighting their significance in reproductive development.

### 3.4. Tumorigenesis 

Malfunction in normal DNA repair processes and genomic stability are major contributors to various types of cancer [87]. Studies that utilize zebrafish models, which allow for the feasible observation of phenotypes at various early time points, have provided valuable insights into this relationship between DNA repair and cancer. Functional investigations in zebrafish have identified several mutants that exhibit tumorigenesis and genomic instability, particularly within the nervous system. One such example is the mutation of *smarcad1a*, frequently observed in patients with malignant peripheral nerve sheath tumors (MPNSTs), a type of cancer originating from Schwann cells, which accelerates tumorigenesis in a zebrafish disease model that has compromised DNA damage repair pathways [88]. Further molecular analysis using *smarcad1a* mutants revealed that DNA damage accumulation was not resolved in the mutant background after ionizing radiation (IR) treatment. These results imply that a compromised DNA DSBR pathway might be responsible for tumorigenesis, a hypothesis that was validated using human MPNST cells.

Notably, defects in MMR genes frequently cause diverse types of cancer, including hereditary colorectal cancer (HNPCC), brain tumors, leukemia, and skin neurofibromas. Homozygous mutations of major zebrafish MMR genes, *mlh1*, *msh2*, and *msh6* have been associated with the development of MPNST and neurofibromas, suggesting a potential role of the MMR system in neuronal tumorigenesis [89]. Additionally, *atm* zebrafish mutants, as previously discussed, do not survive after 12 months of age, mainly due to tumor formation. A loss of atm leads to MPNST-like phenotypes with exophthalmos, as well as several hematologic malignancies with augmentation in monocytes and precursor cells, indicating multiple roles of ATM on tumorigenesis [76]. Another example involves the loss of the tumor suppressor *retinoblastoma1* (*rb1*), a gene responsible for regulating cellular senescence, apoptosis, and DNA repair. Somatic mutations of *rb1* in zebrafish leads to tumor development in the brain and retina regions showing features of neuroectodermal-like and glial-like tumors [90]. Collectively, these findings highlight the pivotal role of zebrafish in studies aimed at understanding the mechanisms underlying cancer development and genomic instability, given its ability to rapidly validate a significant number of candidate cancer driver genes.

### 3.5. Aging

The process of aging is largely attributed to increased genomic instability, which frequently contributes to deficiencies in DNA repair mechanisms [91]. Research on aging using zebrafish has continued to make significant progress, highlighting the importance of genomic integrity in organismal longevity. For instance, the KO of *sirt1*, a gene involved in regulating the cell cycle and senescence, leads to premature aging in adult zebrafish, which is characterized by an increase in ROS, chronic inflammation, and intestinal atrophy [92]. The researchers additionally showed that *sirt1* KO induced apoptotic events, eventually reducing the life span of the zebrafish. Similarly, depletion of the oxidation resistance gene 1 (*oxr1a*), which is responsible for protection against deleterious ROS, leads to emaciation and hypersensitivity against oxidative stress [93]. *oxr1a* mutants showed reduced longevity in zebrafish, demonstrating the importance of DNA repair processes against ROS and DNA oxidative damage for normal aging process.

Telomere shortening is a primary factor in the aging process, and thus defects in telomerase, the enzyme responsible for maintaining telomere length, can accelerate telomere shortening and genomic instability, eventually inducing premature aging [94]. Similarly to humans and mice, zebrafish carrying mutations of the telomerase gene, *tert*, exhibit telomere shortening and chromosome instability [95]. These conditions result in a reduced lifespan and signs of premature aging in adulthood. Recent studies have shown that the specific expression of telomerase within the gut can sufficiently extend the lifespan of these *tert* mutants, demonstrating the tissue-specific effects of *tert* in the aging process [96].

Rag1, a gene crucial for adaptive immunity, also appears to play a role in aging. A loss of *rag1* in zebrafish leads to a shortened lifespan, which is attributed to the accumulation of DNA damage and cellular senescence [97]. Interestingly, *rag1* zebrafish null mutants showed an increased expression of immune-related genes, but a decreased expression of genes related to DNA repair. This observation implied that *rag1* might play a potential role in maintaining genomic integrity to regulate and prevent premature aging. Furthermore, mutations in the genes *atm* and *blm*, which were previously discussed in the context of cancer and germ cell development, also result in decreased lifespans, suggesting that these genes may have multiple functions in the lifespan regulation of living organisms [76,84]. All these studies collectively emphasize the pivotal role of DNA repair mechanisms in aging, illustrating how genomic integrity is closely intertwined with organismal longevity.

## 4. Conclusions

Zebrafish, a viable alternative genetic model to mice, have significantly advanced our understanding of the in vivo functions of DNA repair genes. Investigations focusing on areas such as neurogenesis, hematopoiesis, germ cell development, cancer, and aging have elucidated the profound implications of DNA repair gene mutations on both embryonic and adult processes (Figure 3, Table 2). These discoveries underline the fundamental role of genomic integrity across diverse aspects of vertebrate development. Importantly, the parallels identified between the DNA repair gene functions in zebrafish and humans provide valuable insights, potentially guiding our approach to understanding and managing human diseases associated with DNA repair deficiencies. In summary, zebrafish have proven to be a powerful genetic model system for dissecting the complexities of DNA repair biology and illuminating the in vivo functions of DNA repair pathways in living organisms.

## Figures and Tables

**Figure 1 ijms-24-13120-f001:**
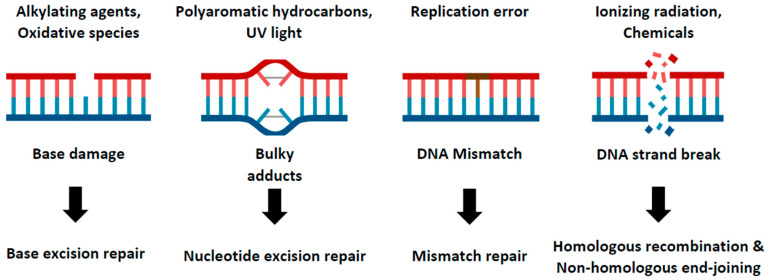
DNA repair pathways in cellular system. This cartoon model illustrates the multiple DNA repair pathways cells have developed to prevent the accumulation of damage. Genomes continuously encounter various kinds of external and internal stresses, which generate diverse forms of DNA damage. Details of these pathways and their response to specific types of DNA damage are described.

**Figure 2 ijms-24-13120-f002:**
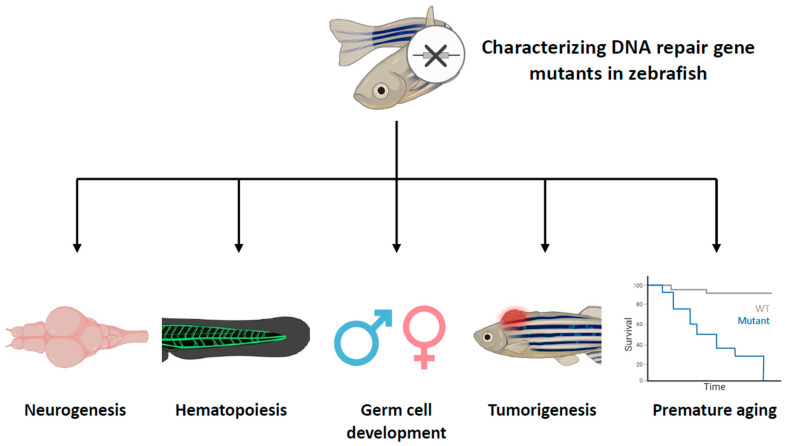
Characterizing DNA repair gene mutants in zebrafish with specific aspects. This figure provides a comprehensive overview of the roles of DNA repair genes in different biological processes in zebrafish. Zebrafish that have mutations in DNA repair genes have been phenotyped, revealing that these genes play diverse roles in various aspects of life processes, including neurogenesis, hematopoiesis, germ cell development, tumorigenesis, and aging. The cross symbol represents the induction of mutations in specific DNA repair genes.

**Figure 3 ijms-24-13120-f003:**
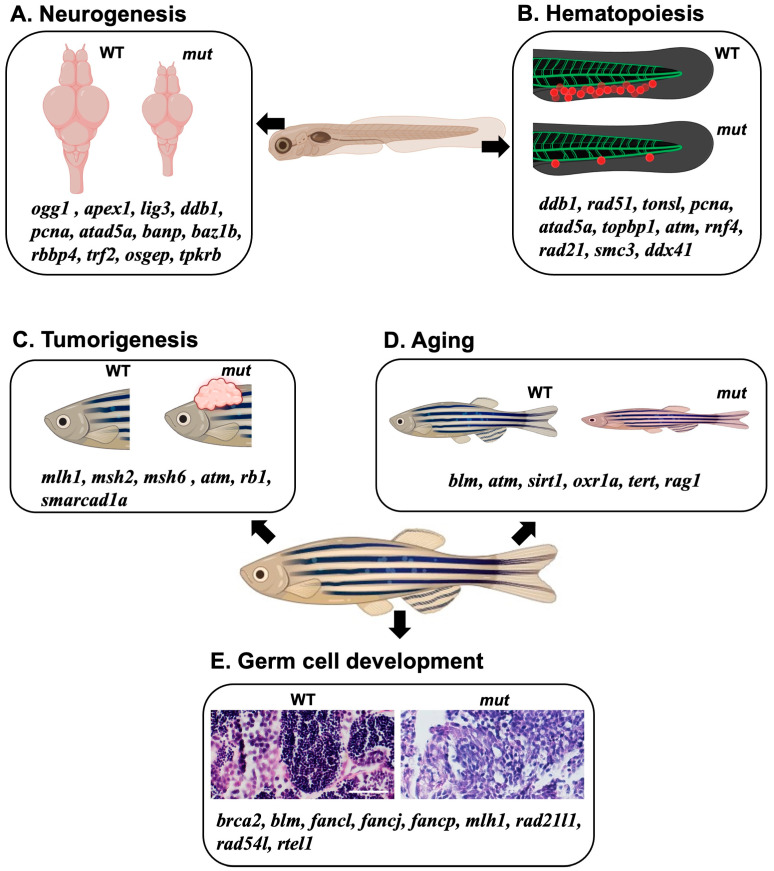
A diagrammatic model summarizing the effects of DNA repair gene KO using a zebrafish model. (**A**,**B**) Collection of DNA repair-related genes that affect neural development and brain formation (**A**) or blood development (**B**) during the embryonic stage when the corresponding genes are knocked out or knocked down. (**C**–**E**) Collection of DNA repair-related genes associated with the phenomena of cancer development (**C**), aging (**D**), and germ cell abnormalities (**E**) that occur during the adult stages when the genes are knocked out. Scale bar = 40 μm.

**Table 2 ijms-24-13120-t002:** Summary. The genes involved in DNA repair processes and the consequences of genetic modification in zebrafish.

Biological Process	Cellular Function	Gene	Consequence of Genetic Modification in Zebrafish
Neurogenesis	BER	*ogg1*	Brain malformation with diminished proliferation and induction of apoptosis
		*apex1*	Abnormality in brain formation with disrupted expression of brain transcription factors
		*lig3*	Immature brain development
	NER	*ddb1*	Malformation of brain and dopaminergic neurons with dysregulation of cell cycle genes
	DNA replication	*pcna*	Reduction in brain size at early stage
		*atad5a*	Abnormal brain morphology with induction of apoptosis
	p53 regulator	*banp*	Defects in eye development with induction of replication stress and p53-dependent apoptosis
	Chromatin remodeler	*baz1b*	Malformation of neural crest cells during embryonic stage and anxiety-associated behavior during adult stage
	Cell cycle regulator	*rbbp4*	Reduced size of brain and eye caused by mitotic arrest of neural progenitor cells
	Telomere protector	*trf2*	Disruption in expression network of neuronal genes
	tRNA processor	*osgep*, *tpkrb*	Microcephalic phenotype with apoptosis induction
Hematopoiesis	NER	*ddb1*	Decreased number of HSPCs with reduction in proliferation
	HR	*rad51*	Reduction in blood population in the kidney marrow of adult zebrafish
	DSBR	*tonsl*	Diminished neutrophil populations during larval stage
	DNA replication	*pcna*, *atad5a*	Decreased number of HSPCs with reduction in proliferation and induction of cell death
	DNA damage checkpoint	*topbp1*	Reduced survival of HSPCs due to the induction of p53-dependent apoptosis
		*atm*	Immune deficiencies from embryonic stage
	Genome maintenance	*rnf4*	Defects in embryonic myeloid development
	Cohesin	*rad21*	Reduction in HSPC marker gene expression
		*smc3*	
	RNA helicase	*ddx41*	Defective differentiation of erythroid progenitors
	Proliferation regulator	*MYCN*	Massive immature HSPCs in kidney marrow with downregulation of BER, HR, and MMR pathways by overexpression
Germ cell development	DNA cross-link repair	*brca2*, *blm*, *fancl*, *fancj*, *fancp*	Female-to-male sex reversal of adult zebrafish
	MMR	*mlh1*	Infertility in male zebrafish with meiotic arrest and aneuploidy of progenies from female zebrafish
	Cohesin	*rad21l1*	Female-to-male sex reversal of adult zebrafish
	HR	*rad54l*	
	Telomere maintenance	*rtel1*	
Tumorigenesis	MMR	*mlh1*, *msh2*, *msh6*	Development of MPNSTs and neurofibromas
	DNA damage checkpoint	*atm*	Formation of MPNSTs and hematologic malignancies
	Cellular senescence	*rb1*	Tumor development in the brain and retina regions
	Genome maintenance	*smarcad1a*	Formation of MPNSTs with compromised DNA damage repair pathways
Aging	MMR	*blm*	Reduced lifespans at adult stages
	DNA damage checkpoint	*atm*	
	Cell cycle regulator	*sirt1*	Premature aging in adult zebrafish with increase in ROS
	ROS response	*oxr1a*	Reduced longevity with emaciation and hypersensitivity against oxidative stress
	Telomere maintenance	*tert*	Premature aging in adulthood with telomere shortening and chromosome instability
	Adaptive immunity	*rag1*	Premature aging with accumulation of DNA damage and cellular senescence

## Data Availability

Not applicable.

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
