# Peer review of "Unraveling DNA Repair Processes In Vivo: Insights from Zebrafish Studies"

_ijms, 2023, doi:10.3390/ijms241713120_

Round 1

Reviewer 1 Report

This is a clear and well-written review focusing on zebrafish as a model for genome stability studies.  The authors have discussed in detail several conditions impacted by defects in DNA repair genes in this organism. One section that is absent is section that discusses the impact of the genes listed in Table I. In mice they are lethal at various embryonic stages but in zebrafish, how do they compare? The addition of this section will make a more compete review.

Author Response

Thank you very much for the comments from Reviewer 1 for our paper. We agree with the reviewer’s point. As Reviewer 1 suggested, we added it as follows on page 3

“ For instance, unlike mice where KO of Ddb1, Atad5, Pcna and Topbp1 leads to early embryonic lethality, zygotic mutations of those genes in zebrafish have allowed them to survive during embryonic stages. Similarly, telo2, mre11 and ino80 KO zebrafish showed viability in the larval stage. Moreover, previous studies have shown that loss of blm, brca2, rad51, palb2, rtel1, and xrcc1 also yields viable zebrafish, even in the adult stages.”

Reviewer 2 Report

The manuscript "Unraveling DNA repair processes in vivo: Insights from zebrafish studies" is well written and undoubtedly of great importance for researchers dealing with DNA repair. The review contains up-to-date information on the effect of DNA repair on the development of the zebrafish organism. In my opinion, the review can be published in the present form. It may be worth including a table listing briefly the genes involved in DNA repair and the consequences of altering their expression. This will increase the volume of the article and duplicate the data described in the text, but it will be very convenient for the reader. I leave this question to the discretion of the authors.

Author Response

We appreciate the comments from Reviewer 2. As suggested by Reviewer 2, we have added a table summarizing the genes involved in DNA repair processes in zebrafish, along with the phenotypes of the mutants. This table can be found as Table 2 on pages 9 to 11.

Reviewer 3 Report

This review is very interesting to read and summarizes the benefit of using zebrafish for DNA repair studies.

I suggest to the authors to add a figure for each point (neurogenesis, hematopoiesis...) to summarize "visually" each players in this pathway.

Author Response

We appreciate the comments from Reviewer 3. Based on the suggestions provided by Reviewer 3, we have created a figure describing the representative effects of DNA repair gene knockout on biological processes, including neurogenesis, hematopoiesis, tumorigenesis, aging, and germ cell development; this figure can be found on page 12. Combined with the newly added Table 2, which summarizes the function of each gene, we believe this figure will visually help readers understand the function of each gene and its relationship to the DNA repair pathway within distinct biological processes.